# Fine-scale genetic structure and rare variant frequencies

**Laurence Gagnon**[1,2], **Claudia Moreau**[1,2], **Catherine Laprise**[1,2,3], **Simon L. Girard**[1,2,4]*

**1** Département des Sciences Fondamentales, Université du Québec à Chicoutimi, Saguenay, Québec, Canada, **2** Centre Intersectoriel en Santé Durable, Université du Québec à Chicoutimi, Saguenay, Québec, Canada, **3** Centre Intégré Universitaire en Santé et Services Sociaux du Saguenay–Lac-Saint-Jean, Saguenay, Québec, Canada, **4** Centre de recherche CERVO, Université Laval, Québec, Québec, Canada

* simon2_girard@uqac.ca

## Abstract

In response to the current challenge in genetic studies to make new associations, we advocate for a shift toward leveraging population fine-scale structure. Our exploration brings to light distinct fine-structure within populations having undergone a founder effect such as the Ashkenazi Jews and the population of the Quebec' province. We leverage the fine-scale population structure to explore its impact on the frequency of rare variants. Notably, we observed an 8-fold increase in frequency for a variant associated with the Usher syndrome in one Quebec subpopulation. Our study underscores that smaller cohorts with greater genetic similarity demonstrate an important increase in rare variant frequencies, offering a promising avenue for new genetic variants' discovery.

## Introduction

Common variants are the primary source of variation identified by genetic association studies. However, despite numerous association analyses conducted in the last years, a significant proportion of the genetic predisposition for many diseases still remains unknown. To address this issue, it is crucial to study rare variants, which often have more significant phenotypic effects, to better understand the "missing heritability". Nevertheless, associations with rare variants present a reduction in statistical power due to the scarcity of individuals carrying these alleles [1].

Several challenges have been encountered in rare variant association analyses, such as accounting for population structure which is an important cofounding factor [2]. Various methods have been proposed to address these challenges and adjust for different confounding effects [2]. Considering this, it is thus essential to explore new creative approaches to overcome all these challenges and facilitate the identification of rare variants.

Therefore, this study aims to use the fine-scale genetic structure of population cohorts as a tool to identify rare variants. We propose that rather than correcting for all confounders, it would be more powerful to seek for rare variants in cohorts with greater genetic similarity. Indeed, given their unique structure, populations that had undergone a founder effect (PFE) have the potential to more readily reveal new genetic associations of rare variants that could

**Data Availability Statement:** The Quebec Regional Reference Sample genotypes are freely available from the institutional repository of BALSAC (DOI: https://doi.org/10.5683/SP3/A5O8DT) and unaffected individuals from the Saguenay-Lac-St-

Jean asthma familial cohort are available at (DOI: https://doi.org/10.5683/SP3/EXHNLH). The Ashkenazi Jews cohort data is available via dbGaP study accession number phs000448.v1.p1, the Himba cohort data is available via dbGaP study accession number phs001995.v1.p1 and the Hutterites cohort data is available via dbGaP study accession number phs001033.v1.p1 (https://www.ncbi.nlm.nih.gov/gap/). The code used for this study can be found in the following GitHub repository: https://github.com/laugag17/world_pop_with_founder_effect.

**Funding:** This work was supported by funding from the Canada Research Chair in Genetics and Genealogy held by SLG. The funders had no role in study design, data collection and analysis, decision to publish, or preparation of the manuscript.

**Competing interests:** The authors have declared that no competing interests exist.

have implications for human health [3–8]. Hence, we will use some of these populations, specifically the Quebec population, Ashkenazi Jews, Himba and Hutterites, to achieve our goals.

## Subjects and methods

This study was approved by the University of Quebec in Chicoutimi (UQAC) Ethics Board. All datasets were accessed on November 11, 2022, and the authors do not have access to any information that could identify individual participants. All methods are summarized in Fig 1.

### Cohorts

The data consist of four different cohorts of PFE. We gained access to data from Quebec, Ashkenazi Jews, Himba and Hutterites (S1 Table in S1 File). We also used the data from the 1000 Genomes Project phase 3 as reference groups from Africa (Mende (MSL)), Europe (British, Northern and Western Europe (GBR and CEU)) and East Asia (Han Chinese and Japanese (CHB and JPT)) as outbred control populations.

### Genotyping data cleaning and imputation

Each individual dataset underwent cleaning using PLINK software v1.9, ensuring individuals with at least 95% genotypes among all SNPs were retained [9]. At the SNP level, we retained SNPs with at least 95% genotypes among all individuals, located on the autosomes and in Hardy–Weinberg equilibrium p > 0.001 (calculated on each cohort).

Subsequently, all datasets were merged (lifting over to hg19 for the Ashkenazi Jews) to retain only common (intersection in all datasets) bi-allelic SNPs. After the merge, individuals with less than 95% genotypes among all SNPs and SNPs with less than 95% genotypes across all individuals were once again filtered out. The final dataset comprises 199,238 SNPs and 4,259 individuals. Related individuals (PLINK pihat > = 0.25) were filtered out to avoid biases in the population structure definition, resulting in a final sample size of 3,683 subjects [10]. This unusually high threshold was applied to retain two populations with high relatedness (first and second degree) (S2 Table in S1 File). The S1 Fig in S1 File demonstrates the low impact of different genetic relatedness thresholds on the pairwise sum of identity-by-descent (IBD) segments length and number. The merged dataset was imputed on TOPMed imputation server, using the reference panel topmed-r2 after lifting over to hg38 [11]. Postimputation quality control filters were applied to remove SNPs with an imputation quality score <0.3 and only biallelic SNPs were kept for further analyses.

### PCA, UMAP and clustering

A Principal Component Analysis (PCA) was done on the merged dataset for visualization and on each individual dataset for clustering using SNPs with a minor allele frequency (MAF) of at least 5%, and after pruning (—indep-pairwise 50 5 0.2) to remove SNPs in linkage disequilibrium.

A Uniform Manifold Approximation and Projection (UMAP) method was performed on the PCA data to investigate the population structure. Pre-processing the data with PCA before performing dimensional reduction with UMAP has been shown to be the most effective computational strategy for exploring population structure within large admixed datasets [12, 13]. UMAP was chosen for its ability to emphasize local data structure while preserving the global structure and for its effectiveness in cluster definition [12, 14, 15]. UMAPs were completed on each individual dataset to form clusters, using the first 5, 4, 7, 8 principal components (PCs) for the Ashkenazi Jews, Quebec, Himba and Hutterites, respectively (S2 and S3

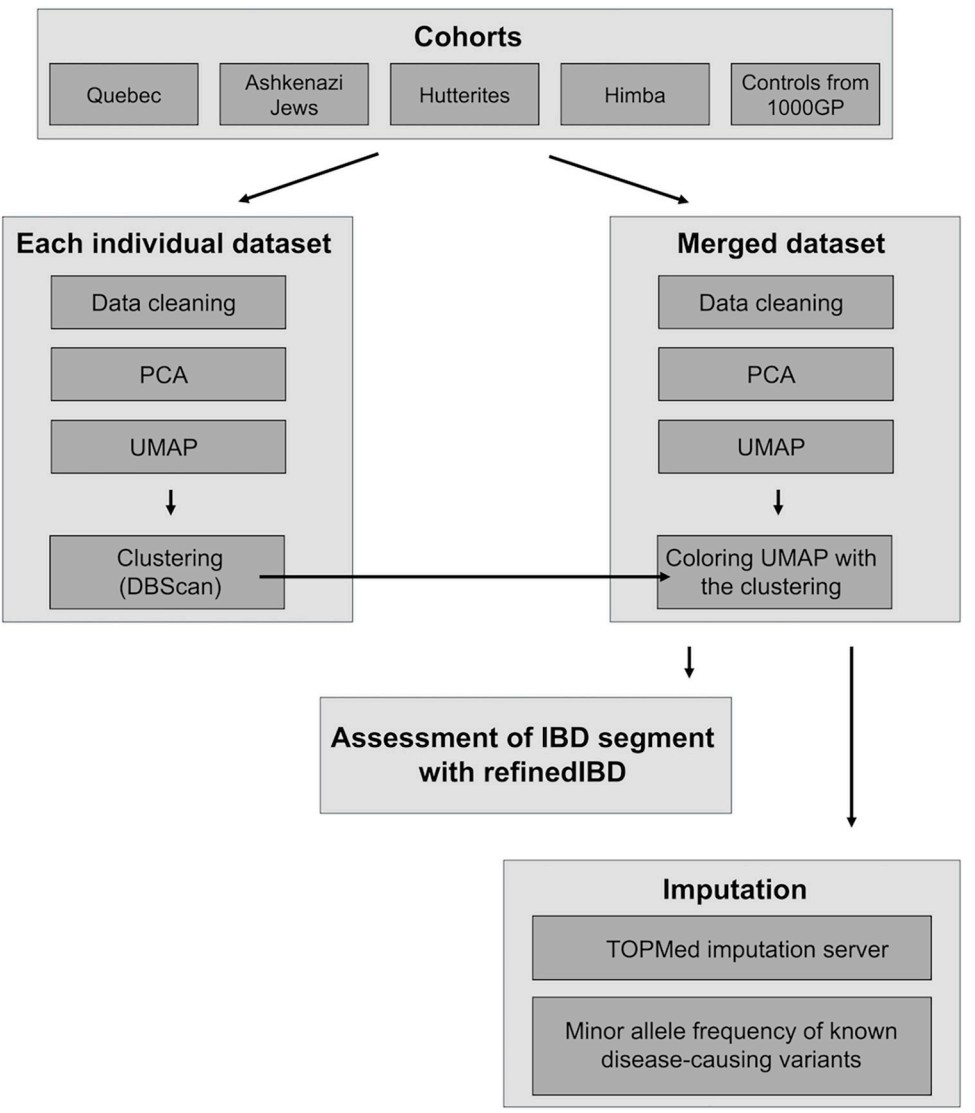

**Fig 1. Simplified view of the flow of analysis.**

Figs in S1 File). The number of PCs was determined by the elbow of the scree plot. Another UMAP was done on the first 8 principal components of the PCA of the merged dataset to visualize the population structure (S4 Fig in S1 File). The UMAPs were realized with the R package "umap" v0.9.2.0 [16]. The n neighbors variable was set to the maximal value (number of individuals in the dataset) (S1 Table in S1 File). The min distance value was set to 0.01 for the UMAPs on each individual dataset to better capture the structure and promote clustering; while it was set to 0.9 for the UMAP on the merged dataset to promote dots splitting and ensure good visualization [17].

The UMAPs of the individual dataset were used to form distinct clusters on each population with the DBScan methods. This method was chosen for its ability in density-based clustering, allowing the capture of clusters with various shapes, including non-convex shapes [18]. The "dbscan" R library v1.1–11 was used for clustering with the minPts parameter set to 4, as the data is two-dimensional [19], and a K-distance graph was done on each dataset to select the

epsilon value from the elbow curve [20]. For the Ashkenazi Jews only, the epsilon value was increased to allow for the formation of a larger cluster that can be seen visually (S3A Fig in S1 File) [20]. The Himba and Hutterites only displayed a single cluster each probably, linked to their lifestyle and reflecting their polygyny and endogamy respectively [21, 22], while Ashkenazi Jews and Quebec exhibited 4 and 5 clusters, respectively (S3 Fig in S1 File). These clusters were afterwards used to color the UMAP of the merge dataset.

## Statistical analysis

Minor allele frequency (MAF) of known disease-causing variants were computed using PLINK software v1.9 on imputed data. The variants were selected for their previously described association to the specific PFE of Quebec (Saguenay–Lac-Saint-Jean (SLSJ) and the Acadians of Gaspe (S5B Fig in S1 File)) and Ashkenazi Jews [23–28]. The MAF was computed both for entire PFE and for the distinct clusters. The MAF of the entire PFE was calculated based on 1000 permutations of the number of individuals in the clusters exhibiting the highest MAF. A p-value was determined by dividing the number of permutations exceeding the MAF of the corresponding cluster by the total number of permutations (1000). The founder variants (listed in Table 1 and S3 Table in S1 File) were selected because of their well-defined association with specific populations. However, limited literature is available concerning specific variants in the Acadians of Gaspe (Quebec-3) which could explain why we found only one variant of interest in this cluster.

## Analysis of IBD segments

The assessment of pairwise IBD segments was performed on the merged dataset of all populations using refinedIBD software v17Jan20 on phased genotypes, which was done using Beagle software version 18May20.d20 [29]. The identified segments were then merged with merge-ibd-segments.17Jan20.102.jar. This software was selected for its robustness and precision in detecting IBD segments [29]. Only segments of 2 cM or more and with a LOD score greater than 3 were retained for further analysis on the level of IBD sharing across the genome.

## Results

### Fine-scale population structure

We characterized the genetic structure of four PFE alongside three reference groups from the 1000 Genomes Project with the aim of using this structure on variants' frequency analysis. A UMAP analysis was conducted to analyze their genetic structure (Fig 2A). The Himba and Hutterites form one tightly packed cluster each, whereas the Ashkenazi Jews and Quebec

**Table 1. Frequency of disease-causing variants known to be associated with a specific population.**

| Population (count) | Associated cluster (count) | Disease | SNP | MAF of the European reference group of 1000GP | Mean MAF of 1000 permutations (P-Value) | MAF of the associated cluster |
|---|---|---|---|---|---|---|
| Ashkenazi Jews (2052) | Ashkenazi Jews-1 (1729) | Familial dysautonomia (ClinVar: 6085) | chr9:108899816: A:G | 0.000 | 0.018 (< 0.001) | 0.021 |
| Quebec (941) | Quebec-2 (233) | Spastic ataxia of Charlevoix-Saguenay (ClinVar: 5512) | chr13:23335031: TA:T | 0.000 | 0.008 (<0.001) | 0.032 |
| Quebec (941) | Quebec-3 (52) | Usher syndrome type I (ClinVar: 5143) | chr11:17531431: C:T | 0.000 | 0.005 (0.002) | 0.039 |

Only the main population and the associated cluster are shown in the table.

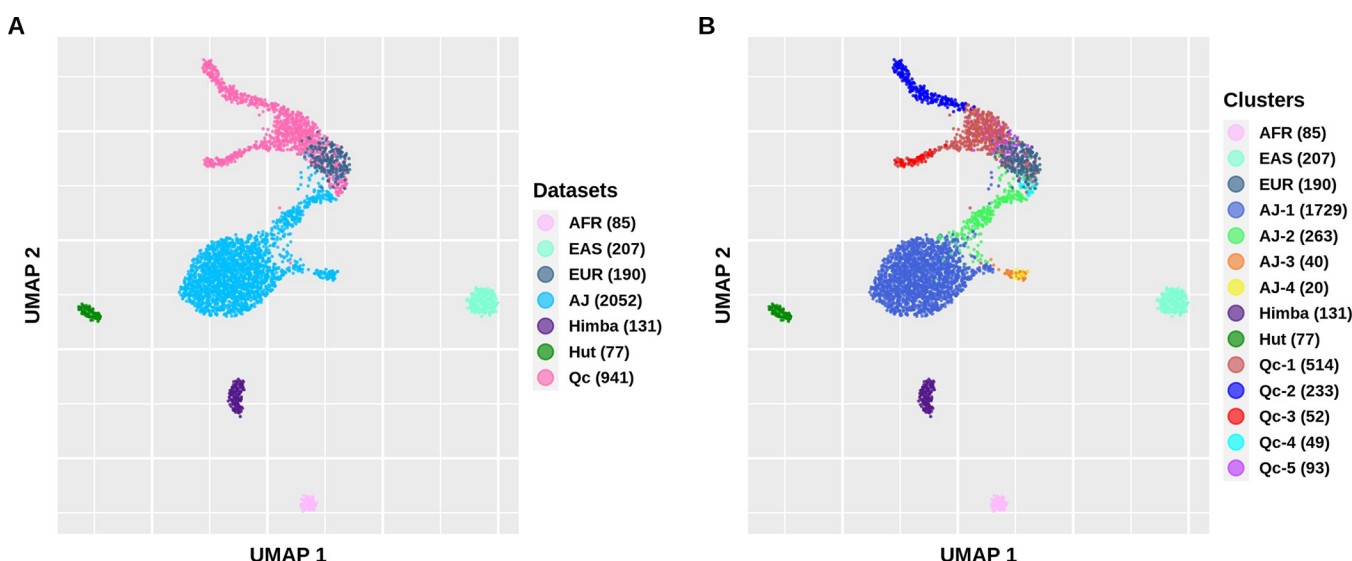

**Fig 2. UMAP of the first 8 principal components of the merged dataset.** Colored according to the origin of the population (A) and the clusters (on S3 Fig in S1 File) (B). AFR African from the 1000 Genomes Project, EAS East Asian from the 1000 Genomes Project, EUR European from the 1000 Genomes Project, AJ Ashkenazi Jews, Hut Hutterites, Qc Quebec.

exhibit a more dispersed pattern and are even interconnected. The Ashkenazi Jews and Quebec went through unique histories of migration, isolation and population expansion. The Ashkenazi Jews-1 cluster possibly represents the Ashkenazi Jewish ancestry, i.e. individuals for whom all four grand-parents were of Ashkenazi Jewish ancestry (S5A Fig in S1 File). In comparison, the Ashkenazi Jews-2 cluster appears to represent a more admixed ancestry, due to its connection to the European reference group and Quebec (Fig 2B). This cluster likely contains individuals with only 1 to 3 of their grand-parents with Ashkenazi Jewish ancestry [7]. Moreover, the analysis of the genetic relatedness among and between clusters reveals that the Ashkenazi Jews clusters are distinct and exhibit greater relatedness within than between clusters (S6A Fig in S1 File). As for Quebec, clusters can be associated with specific ethnocultural groups. Specifically, the Quebec-2 and Quebec-3 represent the Saguenay−Lac-St-Jean (SLSJ) and the Acadians of Gaspe, respectively (S5B Fig in S1 File). These two groups are known for having a genetic structure which distinguishes them from the broader Quebec population that can be associated with Quebec-1 cluster [30–32]. The latter may be related to the initial founder effect in Quebec probably reflecting more diversity amongst founders. This is evident in the lower genetic relatedness observed within the Quebec-1 cluster compared to all the other regional clusters (S6B Fig in S1 File). Undoubtedly, populations like the Ashkenazi Jews and Quebec cannot be treated as single entities due to the presence of fine-structure, even if they were initially perceived as "homogenous populations".

## Impact of using population clusters with higher genetic similarity on variants' frequency

To assess the impact of using the fine-scale population structure on variants' frequency, we looked for variants associated to known founder diseases. These variants were selected for their previously reported genetic association among a specific group. Comparing the frequency of these imputed variants within the associated groups (Quebec-2 for SLSJ, Quebec-3 for Acadians of Gaspe, or Ashkenazi Jews-1 for Ashkenazi Jews) (S5 Fig in S1 File), revealed higher allele frequencies in the specific cluster compared with the other clusters or the whole

population (Table 1). They were also nearly absent from the European reference group. Remarkably, within the Quebec populations, the variants associated with spastic ataxia of Charlevoix-Saguenay and Usher syndrome type I exhibit a 4 (P-Value of 1000 permutation <0.001) and 8 (P-Value of 1000 permutation 0.002) fold increase, respectively, when comparing the specific cluster and the whole population allele frequencies. Notably, this was calculated with a much smaller sample size of 4 and 18-fold, respectively. This trend is also observable while investigating the variant of Familial dysautonomia in the Ashkenazi Jews and other diseases associated with a specific population (Table 1 and S3 Table in S1 File).

## Discussion

This study demonstrates that leveraging fine-scale population structure to intensify the presence of rare variants inside subpopulation clusters could be a promising avenue towards identifying rare variants of potential clinical interest. This approach might also address some challenges related to rare variants' identification. Firstly, rather than controlling for population structure as a confounder [2], we address the impact of population structure directly by using population subdivisions with more similar genetic background. Secondly, concentrating rare variants in smaller cohorts rises their relative frequency, making them less rare and thereby reducing the necessity for huge cohorts and increasing cost-effectiveness for existing cohorts. Therefore, thinking about incorporating fine-scale population structure while designing rare variants' studies can reduce the need for correction algorithms, thereby addressing some of the challenges associated with rare variant associations. This approach could be useful not only in PFE [33], but also in other isolated or even outbred populations since the presence of clusters in more diverse or admixed populations can have striking effect on variants' frequency [34, 35].

Investigating potentially pathogenic variants that are more frequent within targeted populations has the potential to generate positive impacts on public health at the community level and on the discovery of new genes that could be new therapeutic targets. The present study underscores the importance of focusing on smaller populations' fine-scale genetic structure. Indeed, the Acadians in New Brunswick were recently investigated for the first time regarding the frequency of disease-causing variants and the need to study individuals of Acadian ancestry across Canada Atlantic Provinces [24]. Given their unique population structure [30, 31, 36], it would be valuable to investigate the Acadians of Gaspe for potentially pathogenic alleles that may be more frequent in this population. In fact, despite increasing evidence of a strong and recent founder effect in the Gaspesian population of Quebec, which may have led to notable local changes in allele frequencies, this regional population has been relatively understudied [30]. In contrast, other regional populations of Quebec, such as the SLSJ, and also the Ashkenazi Jews have been the focus of many studies for rare genetic variants that have increased in frequency due to the founder effect [6, 7, 28, 37–39]. Even then, revisiting these populations using fine-structure in population cohorts, might reveal some yet undiscovered rare variants that are more frequent in these specific groups.

As for Hutterites and Himba, they can be studied as a whole as they do not subdivide into clusters. Indeed, Hutterites are known to practice endogamy and live in community and Himba individuals were documented to practice polygyny and live in a pastoralist way [21, 22]. This way of living promotes very close links between individuals and could explain the absence of population subdivision at the level tested. This is also evident in their proportion of pairs sharing an IBD segment across the genome, which reaches the highest level among all PFE (S4 Table in S1 File). However, the Hutterites have been extensively studied for more prevalent diseases within the population [40, 41], whereas the Himba have not, despite their

unique history and the precedence of increased homozygosity [42]. Our findings suggest that the Himba should be a center of interest in rare variant studies. This highlights the issue that genetic studies are predominantly focused on populations of European ancestry [43–45]. As a result, there is a lack of understanding of the genetic diversity present in other populations, which could harbor rare variants of significant importance. By shifting the focus to fine-scale genetic structures—whether analyzing entire populations or their subdivisions—researchers could improve the identification of rare variants by concentrating on individuals with more similar genetic backgrounds. This approach not only enhances our understanding of genetic diversity across different populations but also opens new avenues for discovering rare variants that may contribute to human health and disease.

There are some limitations present in this work. UMAP tends to prioritize local distances, which means that points that are close together in the projection are likely to be similar in the dataset. However, points that are farther apart might not actually be very different [12]. Nevertheless, this allowed us to identify and easily select clusters of very similar individuals. We acknowledge that this method cannot visually represent genetic variation on a continuum, as previously questioned and discussed [46]. Furthermore, the use of imputed data might have led to the loss of some rare variants resulting in an underestimation of the frequency of certain rare variants. Despite this, we were able to demonstrate an increase in frequency through population clusters, although the number of variants tested was likely lower, as evidenced by the detection of only one variant in the Acadians of Gaspe.

In conclusion, we suggest a novel approach, parallel to the already existing strategies, that leverages the fine-scale genetic structure of a population cohort to refine analysis models for rare variant studies. This cost-effectiveness strategy would help to enhance the value of existing large cohorts. We believe that cohorts composed of fewer individuals with a common genetic background would help in discovering new rare genetic associations, as they would be easier to find given their increased frequency.

## Supporting information

**S1 File. Includes S1-S6 Figs and S1-S4 Tables.**
(DOCX)

## Acknowledgments

This work was made possible by the Digital Research Alliance of Canada which provided access to storage and computing resources. We are extremely grateful to all participants of this research. We would like to thank Hélène Vézina, Damian Labuda and their team for the Quebec Regional Reference Sample cohort constitution. LG received scholarship from the Fonds de recherche du Québec—Santé and the Canadian Institutes of Health Research. CL is the director of the Centre Intersectoriel en Santé Durable (http://www.uqac.ca/santedurable), the chairholder of the Canada Research Chair in the Genomics of asthma and allergic diseases (http://www.chairs.gc.ca) and co-holder of the Chaire en santé durable du Québec (http://www.chairesantedurable.ca).

## Author Contributions

**Conceptualization:** Laurence Gagnon, Claudia Moreau, Simon L. Girard.

**Data curation:** Laurence Gagnon.

**Formal analysis:** Laurence Gagnon.

**Funding acquisition:** Simon L. Girard.

**Investigation:** Laurence Gagnon.

**Methodology:** Laurence Gagnon, Claudia Moreau.

**Resources:** Catherine Laprise.

**Supervision:** Claudia Moreau, Simon L. Girard.

**Writing – original draft:** Laurence Gagnon.

**Writing – review & editing:** Claudia Moreau, Catherine Laprise, Simon L. Girard.

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
