## [Decision Letter · Decision Letter 0]

16 Jul 2024

PONE-D-24-18509Fine-scale genetic structure and rare variant frequenciesPLOS ONE

Dear Dr. Girard,

Thank you for submitting your manuscript to PLOS ONE. After careful consideration, we feel that it has merit but does not fully meet PLOS ONE’s publication criteria as it currently stands. Therefore, we invite you to submit a revised version of the manuscript that addresses the points raised during the review process.

We look forward to receiving your revised manuscript.

Kind regards,

Nejat Mahdieh

Academic Editor

PLOS ONE

 [This work was supported by funding from the Canada Research Chair in Genetics and Genealogy hold by SLG.].  

Additional Editor Comments (if provided):

Reviewers' comments:

Reviewer's Responses to Questions

**Comments to the Author**

1. Is the manuscript technically sound, and do the data support the conclusions?

Reviewer #1: Partly

Reviewer #2: Partly

2. Has the statistical analysis been performed appropriately and rigorously? 

Reviewer #1: No

Reviewer #2: Yes

3. Have the authors made all data underlying the findings in their manuscript fully available?

Reviewer #1: Yes

Reviewer #2: Yes

4. Is the manuscript presented in an intelligible fashion and written in standard English?

Reviewer #1: Yes

Reviewer #2: Yes

5. Review Comments to the Author

Reviewer #1: The primary objective of this study was to propose an approach on identifying fine-scale genetic structure of populations that can be used for discovering new rare genetic variants in association analyses. While the topic is interesting, the authors need to address several major concerns and include more detail on their method and analyses to strengthen their manuscript. Please see the rest of the review in the attachment.

Reviewer #2: My suggestions are as follows:

The abstract could benefit from more specific results, such as key findings and their implications, to provide a clearer snapshot of the study's outcomes.

Lacks mention of specific populations studied or the methods used.

The introduction is quite general and could be more focused on specific hypotheses or research questions the study aims to address.

The importance of fine-scale structure is mentioned, but the introduction does not sufficiently elaborate on how this study uniquely contributes to this field.

The explanation of statistical methods is somewhat complex and may benefit from simplification or clarification for a broader audience.

The rationale behind the choice of the minPts parameter for DBScan is not thoroughly explained.

Potential limitations of using imputed data for MAF calculations are acknowledged but not deeply discussed in terms of their impact on study results.

The results section is descriptive but lacks a clear connection to broader implications or potential applications of the findings.

Some findings, like the single variant in Quebec-3, are mentioned briefly without in-depth exploration of their significance.

The presence of fine-structure in populations is demonstrated, but the narrative could better connect these findings to the overarching goal of addressing "missing heritability."

The discussion reiterates the findings without providing enough critical analysis or addressing potential limitations of the study.

The broader impact of these findings on the field of genetics or potential future research directions is not thoroughly discussed.

Could benefit from more specific examples of how this approach could be applied in practical scenarios or public health contexts.

The manuscript could benefit from a clearer narrative thread connecting the introduction, methods, results, and discussion.

Greater emphasis on the practical implications and future directions of this research would strengthen the manuscript.

Some sections, particularly the methods and results, may be overly technical for a broad audience and could be simplified or clarified.

By addressing these drawbacks, the manuscript would be more compelling and accessible to a wider audience, enhancing its contribution to the field.

6. PLOS authors have the option to publish the peer review history of their article (what does this mean?). If published, this will include your full peer review and any attached files.

Reviewer #1: No

Reviewer #2: No

---

## [Author Response · Author response to Decision Letter 0]

30 Sep 2024

Response to Reviewers

We thank the reviewers for their time and useful comments. We clarified the objectives and the broader implications in the field and added the supplementary methods in the main text to make it clearer. The discussion has been rewritten for clarity and better alignment with the project aims. We have added a new Fig. 1 and supplementary Fig. 2 and 4. In addition, we provide a point-by-point response to every concern raised by both reviewers. Line numbers are reported on the track changes version. Some reviewer comments were grouped.

Reviewer #1 : 

The primary objective of this study was to propose an approach on identifying fine-scale genetic structure of populations that can be used for discovering new rare genetic variants in association analyses. While the topic is interesting, the authors need to address several major concerns and include more detail on their method and analyses to strengthen their manuscript.

1. The authors did not discuss the current approaches, e.g., fineSTRUCTURE, ipPCA, etc., for identifying fine-scale population structure. It would be helpful to compare these approaches to their method, clearly state the novelty of their method, and apply the current approaches to the datasets explored in the manuscript to see whether fine structure can be captured within Ashkenazi Jews and the Quebec populations. The strengths/weaknesses of their method should be provided in the Discussion section. 

We explained the choice and novelty of our method in the article at line 100 to 105. The focus of our study was not to present a new method for identifying fine-structure, but to use the fine-structure as a tool to concentrate and identify new rare variants that become more frequent in the small clusters. We rewrote parts of the introduction and discussion to make the objectives and implications clearer in the text. We acknowledge that other methods could be used to represent population fine-structure. However, we used the recent method described in (Diaz-Papkovich et al. 2019) that was shown to be computationally efficient and able to reveal fine-scale genetic structure while preserving the global structure (Diaz-Papkovich et al. 2019, 2021, 2023; McConville et al. 2021). Moreover, this method does not rely on source populations such as fineSTRUCTURE. We also add a new limitation paragraph of the discussion where we talk about UMAP.

2. In the Supplementary section, the authors stated that they used a threshold of “pihat >= 0.25” for filtering. Why did the authors choose to remove genetically-related individuals in their analysis?

3. Despite using a threshold of “pihat >=0.25”, it’s possible that the final dataset includes individuals with third-degree relatedness. Applying principal component analysis (PCA) to this is not appropriate as PCA is not robust to the presence of genetic relatedness, resulting in distortions in the overall PCA pattern. The authors need to use an adjusted PCA method, such as PC-AiR (Conomos et al., Genet Epidemiol., 2015), which appropriately accounts for genetic relatedness.

We removed close relatedness because they can bias population genetics and structure analyses (Wang 2018). Wang therein suggests removing close relatives (first and second degree) before doing population genetic analysis. It has been added in the text (line 84-87). 

We also acknowledge that a pihat threshold of >= 0.25 will include third-degree related individuals. Consequently, we tried with many different thresholds ensuring that the structure separating the African, Asian and European was kept, no matter the value of the pihat. The results are shown in the PCAs below. With this analysis and the article of Wang 2018, we decided to choose pihat 0.25 to retain two populations, the Hutterites and the Himba, which would have been eliminated due to the low number of individuals (S2 Table).

4. The authors should provide a figure showing plots of the principal components (PCs). 

This was added as S2 and S4 Fig in the Supporting information.

5. The authors chose to use UMAP to visualize the population structure and labeled the points with the corresponding ethnicity and ancestry. As depicted in Figure 1 of the manuscript, UMAP tends to exaggerate the distinctiveness of populations, failing to show that genetic variation is a continuum. This issue was discussed recently in regard to the Nature paper, “Genomic data in the All of Us Research Program” (see link for discussion). Can the authors consider alternatives to UMAP for their approach? If not, the authors should at least state in the Discussion section the shortcomings of UMAP and their approach

We thank the reviewer for the very interesting article and discussion about the All of Us paper. Although, we chose to use a UMAP precisely for its ability to find finer features hidden in the main population structure (Diaz-Papkovich et al. 2019, 2021). This method is also easy to use, fast, and works well with big datasets and with individuals of admixed background. In this analysis, our goal was to identify clearly distinct clusters, which justified our decision to use UMAP. This justification was added in the text at line 100 to 105 and 255 to 260. 

6. How did the authors decide on using the number of PCs for performing UMAP? Please provide more detail on this in the Results section.

The number of PCs selected to make the UMAP of the individual dataset was chosen based on the elbow point of the scree plot. The first 5, 4, 7, 8 PCs were selected for the Ashkenazi Jews, Quebec, Himba and Hutterites, respectively. This has been specified in the text at line 105 to 108. 

7. Why did the authors choose specific populations within the African, East Asian, and European superpopulations for their analysis ? In addition, the authors need to clearly define all population labels, i.e., AFR, EAS, EUR, etc., in Figure 1. 

We chose these specific populations because they are outbred with well-defined ancestry. We wanted to make sure that the studied populations with founder effects were positioned where we expected them to be and that there was no bias or problem with the data. This was added in the text line 72.

The labels are defined in the caption of the figure 2, line 177 to 181.

8. More detail should be provided on the DBScan method. Specifically, how did the authors tune the parameters, e.g., the epsilon value and “minPts parameter set to 4”, for this clustering approach? Did the authors consider other flexible clustering approaches that can “capture…clusters with various shapes” besides DBScan? 

Reviewer #2: The rationale behind the choice of the minPts parameter for DBScan is not thoroughly explained. 

For the DBScan method, the minPts parameter needs to be set to 4 for two dimensions’ data (Ester et al. 1996). A K-distance graphic was done to get the epsilon value from the elbow of the curve (Sander et al. 1998). Only for the Ashkenazi Jews, the epsilon value was set higher to enable the creation of a larger cluster, as it could be seen visually from the graphic (S3A Fig) (Sander et al. 1998). This was added to the text line 119 to 122.

We tried other clustering methods such as K-means, the spectral clustering, the gaussian mixture model, the ward hierarchical clustering and OPTICS. We finally chose DBScan because it was a density-based clustering that created clusters that could make sense with the Quebec population structure that was already known (Roy-Gagnon et al. 2011; Gauvin et al. 2014; Gagnon et al. 2024). 

9. More detail needs to be provided on how the authors selected “known founder disease variants.” What method was used for the association analysis? Did the authors adjust for any confounders in their analysis? Given that the dataset includes genetically related individuals (third-degree relatedness), was this adjusted for in the association analysis? 

The variants were selected for their previously described association in the literature to the specific PFE of Quebec (Saguenay-Lac-Saint-Jean (SLSJ) and the Acadians of Gaspe) and Ashkenazi Jews (Charrow 2004; Ebermann et al. 2007; Robichaud et al. 2022; Wallace and Bean 2022; ARUP Consult 2023; Cruz Marino et al. 2023). It was added in the text at line 130-132.

We did not perform a true genetic association analysis in this article, as we do not have any phenotype. We rather explored the concentrating effect of using population clusters on rare variants associated with a specific population. This method could enhance the ability to detect genetic associations with previously published variants and could also help to identify new variants that may be more common in specific populations but have not yet been discovered. This was clarified in the whole text.

10. In Table 1, the authors showed a p-value of 0, which is incorrect. Please provide the p-value in scientific notation. Furthermore, the authors should include the effect size and the 95% confidence intervals from their association analysis. 

As stated above, we did not perform an association analysis, but rather a permutation test to ensure that the observed scenario would not happen by chance. However, the reviewer is right that the p-value cannot be 0, this was corrected in the article. Since we performed 1000 permutations, when the observed scenario never happens, we set the p-value < 0.001.

Review #2 : 

The abstract could benefit from more specific results, such as key findings and their implications, to provide a clearer snapshot of the study's outcomes. Lacks mention of specific populations studied or the methods used. 

More details were added in the abstract.

The introduction is quite general and could be more focused on specific hypotheses or research questions the study aims to address.

The importance of fine-scale structure is mentioned, but the introduction does not sufficiently elaborate on how this study uniquely contributes to this field. 

More details were added in the introduction and on the specific objectives of the study.

The explanation of statistical methods is somewhat complex and may benefit from simplification or clarification for a broader audience. 

Some sections, particularly the methods and results, may be overly technical for a broad audience and could be simplified or clarified. 

We have contradictory requests from both reviewers in this specific case. To clarify and improve readability, we have integrated the supplementary methods into the main text. Additionally, to enhance the clarity of the methods section for a broader audience, we have included Figure 1, which summarizes the flow of the analysis.

Potential limitations of using imputed data for MAF calculations are acknowledged but not deeply discussed in terms of their impact on study results. 

This was added in the new limitation section, line 255 to 264.

The results section is descriptive but lacks a clear connection to broader implications or potential applications of the findings. 

The result section was reworked to make it more clear with the aims of the study. Part of the text was moved to the discussion section.

Some findings, like the single variant in Quebec-3, are mentioned briefly without in-depth exploration of their significance.

The presence of fine-structure in populations is demonstrated, but the narrative could better connect these findings to the overarching goal of addressing "missing heritability." 

This was added to the discussion 221-234.

The discussion reiterates the findings without providing enough critical analysis or addressing potential limitations of the study. 

The discussion section was rewritten. We also added a limitation paragraph in the discussion section.

The broader impact of these findings on the field of genetics or potential future research directions is not thoroughly discussed.

Could benefit from more specific examples of how this approach could be applied in practical scenarios or public health contexts.

Greater emphasis on the practical implications and future directions of this research would strengthen the manuscript.

We added more details on implications and future directions in the discussion. It is also worth to note that we have another paper in preparation in the lab that uses a similar approach to show previously unreported founder variants in Quebec and a greater deleterious mutational burden in individuals from Quebec compared to Europeans. This new paper also highlights the potential underestimation of rare disease prevalence.

The manuscript could benefit from a clearer narrative thread connecting the introduction, methods, results, and discussion.

The text was reworked to make it clearer.

References :

ARUP Consult (2023) Ashkenazi Jewish Genetic Diseases Panel. In: Ashkenazi Jewish Genetic Diseases Panel. https://arupconsult.com/ati/ashkenazi-jewish-genetic-diseases-panel#clinical-sensitivity

Charrow J (2004) Ashkenazi Jewish genetic disorders. Fam Cancer 3:201–206. https://doi.org/10.1007/s10689-004-9545-z

Cruz Marino T, Leblanc J, Pratte A, et al (2023) Portrait of autosomal recessive diseases in the French-Canadian founder population of Saguenay-Lac-Saint-Jean. Am J Med Genet A 191:1145–1163. https://doi.org/10.1002/ajmg.a.63147

Diaz-Papkovich A, Anderson-Trocmé L, Ben-Eghan C, Gravel S (2019) UMAP reveals cryptic population structure and phenotype heterogeneity in large genomic cohorts. PLOS Genetics 15:e1008432. https://doi.org/10.1371/journal.pgen.1008432

Diaz-Papkovich A, Anderson-Trocmé L, Gravel S (2021) A review of UMAP in population genetics. J Hum Genet 66:85–91. https://doi.org/10.1038/s10038-020-00851-4

Diaz-Papkovich A, Zabad S, Ben-Eghan C, et al (2023) Topological stratification of continuous genetic variation in large biobanks. 2023.07.06.548007

Ebermann I, Lopez I, Bitner-Glindzicz M, et al (2007) Deafblindness in French Canadians from Quebec: a predominant founder mutation in the USH1C gene provides the first genetic link with the Acadian population. Genome Biol 8:R47. https://doi.org/10.1186/gb-2007-8-4-r47

Ester M, Kriegel H-P, Sander J, Xu X (1996) A density-based algorithm for discovering clusters in large spatial databases with noise. pp 226–231

Gagnon L, Moreau C, Laprise C, et al (2024) Deciphering the genetic structure of the Quebec founder population using genealogies. European Journal of Human Genetics 32:91–97. https://doi.org/10.1038/s41431-023-01356-2

Gauvin H, Moreau C, Lefebvre J-F, et al (2014) Genome-wide patterns of identity-by-descent sharing in the French Canadian founder population. Eur J Hum Genet 22:814–821. https://doi.org/10.1038/ejhg.2013.227

McConville, Santos-Rodríguez, Piechocki, Craddock (2021) N2D: (Not Too) Deep Clustering via Clustering the Local Manifold of an Autoencoded Embedding. In: 2020 25th International Conference on Pattern Recognition (ICPR). pp 5145–5152

Robichaud PP, Allain EP, Belbraouet S, et al (2022) Pathogenic variants carrier screening in New Brunswick: Acadians reveal high carrier frequency for multiple genetic disorders. BMC Medical Genomics 15:98. https://doi.org/10.1186/s12920-022-01249-1

Roy-Gagnon MH, Moreau C, Bherer C, et al (2011) Genomic and genealogical investigation of the French Canadian founder population structure. Human Genetics 129:521–531. https://doi.org/10.1007/s00439-010-0945-x

Sander J, Ester M, Kriegel H-P, Xu X (1998) Density-based clustering in spatial databases: The algorithm gdbscan and its applications. Data mining and knowledge discovery 2:169–194

Wallace SE, Bean LJ (2022) Resources for Genetics Professionals — Genetic Disorders Associated with Founder Variants Common in the Hutterite Population. In: GeneReviews. University of Washington, Seattle

Wang J (2018) Effects of sampling close relatives on some elementary population genetics analyses. Mol Ecol Resour 18:41–54. https://doi.org/10.1111/1755-0998.12708

---

## [Decision Letter · Decision Letter 1]

21 Oct 2024

Fine-scale genetic structure and rare variant frequencies

PONE-D-24-18509R1

Dear Dr. Girard,

We’re pleased to inform you that your manuscript has been judged scientifically suitable for publication and will be formally accepted for publication once it meets all outstanding technical requirements.

Kind regards,

Nejat Mahdieh

Academic Editor

PLOS ONE

Additional Editor Comments (optional):

Reviewers' comments:

Reviewer's Responses to Questions

**Comments to the Author**

1. If the authors have adequately addressed your comments raised in a previous round of review and you feel that this manuscript is now acceptable for publication, you may indicate that here to bypass the “Comments to the Author” section, enter your conflict of interest statement in the “Confidential to Editor” section, and submit your "Accept" recommendation.

Reviewer #1: All comments have been addressed

2. Is the manuscript technically sound, and do the data support the conclusions?

Reviewer #1: Yes

3. Has the statistical analysis been performed appropriately and rigorously? 

Reviewer #1: Yes

4. Have the authors made all data underlying the findings in their manuscript fully available?

Reviewer #1: Yes

5. Is the manuscript presented in an intelligible fashion and written in standard English?

Reviewer #1: Yes

6. Review Comments to the Author

Reviewer #1: The authors addressed the comments and feedback appropriately, and I have no further requests or suggestions.

7. PLOS authors have the option to publish the peer review history of their article (what does this mean?). If published, this will include your full peer review and any attached files.

Reviewer #1: No

---

## [Editor Report · Acceptance letter]

25 Oct 2024

PONE-D-24-18509R1 

PLOS ONE

Dear Dr. Girard, 

I'm pleased to inform you that your manuscript has been deemed suitable for publication in PLOS ONE. Congratulations! Your manuscript is now being handed over to our production team.

Kind regards, 

on behalf of

Dr. Nejat Mahdieh 

Academic Editor

PLOS ONE